# Academic Competitions

**Hugo Jair Escalante**                                    HUGOJAIR@INAOEP.MX
*Instituto Nacional de Astrofísica,*
*Óptica y Electrónica,*
*Tonantzintla, 72840, Puebla, Mexico*
*The University of Texas at El Paso,*
*500 W University Ave, El Paso, TX 79968*

**Aleksandra Kruchinina**         ALEKSANDRA.KRUCHININA@UNIVERSITE-PARIS-SACLAY.FR
*Université Paris Saclay*
*Paris, France*

**Reviewed on OpenReview:** *https: // openreview. net/ forum? id= Oc1taS2iYd*

## Abstract

Competitions comprise effective means for (i) advancing the state of the art, (ii) putting in the spotlight of a scientific community specific topics and problems, as well as (iii) closing the gap for under represented communities in terms of accessing and participating in the shaping of research fields. Competitions can be traced back for centuries and their achievements have had great influence in our modern world. Recently, they (re)gained popularity, with the overwhelming amounts of data that is being generated in different domains, as well as the need of pushing the barriers of existing methods, and available tools to handle such data. This chapter provides a survey of academic challenges in the context of machine learning and related fields. We review the most influential competitions in the last few years and analyze challenges per area of knowledge. The aims of scientific challenges, their goals, major achievements and expectations for the next few years are reviewed. An associated repository is available here: https://hugojair.github.io/challenges-survey/

**Keywords:**   Academic competitions and challenges, Survey of academic challenges, Impact of academic competitions.

## 1 Introduction

Competitions are nowadays a key component of academic events, as they comprise effective means for making rapid progress in specific topics. By posing a challenge to the academic community, competition organizers contribute to pushing the state of the art in specific subjects and/or to solve problems of practical importance. In fact, challenges are a channel for the reproducibility and validation of experimental results in specific scenarios and tasks.

We can distinguish two types of competitions: those associated to industry or aiming at solving a practical problem, and those that are associated to a research question (academic competitions). While sometimes it is difficult to typecast competitions in these two categories, one can often identify a tendency to either variant. This chapter focuses on academic competitions, although some of the reviewed challenges are often associated to industry too. An academic competition can be defined as a *contest that aims to answer a scientific question via crowd sourcing where participants propose innovative solutions, ideally the challenge will push the state-of-the-art and have a long-lasting impact and/or an*

*established benchmark.* In this context, academic competitions relying on data have been organized for a while in a number of fields like natural language processing (Harman, 1993), machine learning (Guyon et al., 2004) and knowledge discovery in databases[1], however, their spread and impact has considerably increased during the last decade, see Figure 1 for statistics of the CodaLab platform (Pavao et al., 2023).

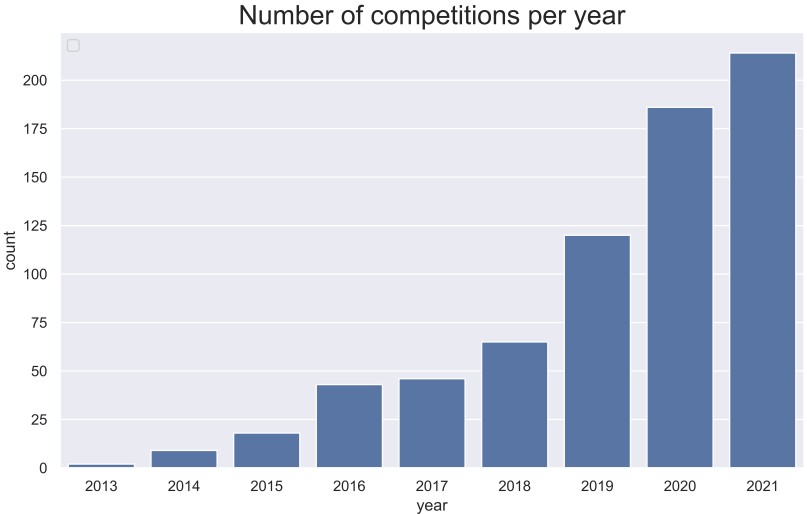

Figure 1: Evolution of the number of competitions each year. Data gathered from *Co-daLab Competitions* (Pavao et al., 2023), a platform with a community focused on academic competitions recognized consecutive years as the platform with most organized competitions (Carlens, 2023, 2024, 2025).

As a consequence of this growth, we can witness the permeation and influence that competitions have had in a number of fields. This chapter aims to survey academic competitions and their impact in the last few years. The objective is to provide the reader with a snapshot of the rise and establishment of academic competitions, and to outline open questions that could be addressed with support of contests in the near future. We have focused on machine learning competitions with emphasis on academic challenges. Nevertheless, competitions from other related fields are also briefly reviewed.

The remainder of this chapter is organized as follows. Next section provides a brief historical review of competitions in the context of academia and their impact in different fields. Then, in Section 3, we review academic competitions in terms of the associated field. Section 4 elaborates on ongoing trends in the era of foundational models. Finally in Section 5 we outline some thoughts and ideas on the future of academic competitions.

---

1. https://www.kdd.org/kdd-cup/view/kdd-cup-1997

## 2 A review of academic challenges: past and present

This section provides a survey on academic challenges in the context of machine learning and related fields.

### 2.1 Historical review

While it is a daunting task to provide a comprehensive timeline of the evolution of challenges in machine learning and related fields, this section aims at providing a generic overview. Perhaps the first memorable *challenge* is the Longitude Act issued in 1714. It asked participants to develop a method to determine longitude up to a half degree accuracy (i.e., about 69 miles in distance if one is placed in the Meridian). After years of milestones and fierce competition, Thomas Harrison was acknowledged as the winner of this *challenge*. The main incentive, in addition to scientific curiosity, was a monetary prize offered by the British crown that today would be equivalent to millions of pounds.

This form of incentive has guided several other competitions organized by governments[2], for example the DARPA (Defense Advanced Research Projects Agency) grand challenge[3] series that for years organized competitions for building an all-terrain autonomous vehicle. These type of challenges are still being organized nowadays, not only by governments but also by other institutions and even the private sector. Consider for instance the funded challenges organized by the National Institute of Standards and Technology[4] (NIST) and the latest editions of the X-Prize Challenge[5] and the Longitude Prize[6], both targeting critical health problems via challenges in their most recent editions. This same model of making progress via crowd sourcing has been adopted by academy for a while now. The first efforts in this direction arose in the 90s, it was in that decade that the first RoboCup, ICDAR (International Conference on Document Analysis and Recognition), KDD Cup (Knowledge Discovery and Data Mining Tools Competition) and TREC (Text Retrieval Conference) competitions were organized. Such challenges are still being organized on a yearly basis, and they have helped to guide the progress in their respective fields.

RoboCup initially focused on the development of robotic systems able to eventually *play* Soccer at human level (Kitano et al., 1998). With currently more than 25 editions, RoboCup has evolved in the type of tasks addressed in the context of the challenge. For instance, the 2026 edition[7] comprises leagues on rescue robots, service robots, soccer playing robots, industrial robots and even a junior league for kids, where each league has multiple tracks. RoboCup competition model has motivated progress on different sub fields within robotics, from hardware to robot control and multi agent communication among others, see (Visser, 2016) for a survey on the achievements of this first 20 editions of RoboCup. Together with the DARPA challenge, RoboCup has largely guided the progress of autonomous robotic agents that interact in physical environments.

---

2. `https://www.nasa.gov/solve/history-of-challenges`

3. `https://www.darpa.mil/news-events/2014-03-13`

4. `https://www.nist.gov/`

5. `https://www.xprize.org/challenges`

6. `https://longitudeprize.org/`

7. `https://2022.robocup.org/`

Organized by NIST, TREC is another of the *long-lived* evaluation forums that arose in the early 90s (Harman, 1993). TREC initially focused on text retrieval tasks. Unlike RoboCup, where solutions were tested lively during the event, TREC asked participants to submit *runs* of their retrieval systems in response to a series of queries. By that time this represented a great opportunity for participants to evaluate their solutions in large scale and realistic retrieval scenarios. This evaluation model actually is still popular among text-based evaluation forums (see e.g., SemEval[8]). The TREC forum has evolved and now it focuses on a diversity of tasks around information retrieval (e.g., retrieval of clinical treatments based on patients' cases). Additionally, TREC gave rise to a number of efforts like CLEF (Conference and Labs of the Evaluation Forum), ImageCLEF and TRECVID. They split from TREC to deal with specific sub problems such as: question answering, image and video retrieval, respectively.

In terms of OCR, there were also efforts aiming to boost research in this open problem during the 90s (Garris et al., 1997). The first ICDAR conference took place in 1991, although well documented competitions started in the early 00s (see, e.g., (Lucas et al., 2003)), it seems that competitions associated to digital document analysis were associated to ICDAR since the early 90s, see (Matsui et al., 1993). By that time, NIST released a large dataset of handwritten digits (Grother, 1995) with detailed instructions on preprocessing, evaluation protocols and reference results. While this was not precisely an academic competition, this effort allowed reproducibility in times where the world was starting to benefit from information spread throughout the internet. The impact of this effort has been such that, in addition to motivating breakthroughs in OCR, established the MNIST benchmark as a reference problem for supervised learning (see e.g., Yann Lecun's site[9] on results in a subset of this benchmark). Please note that MNIST is considered a *biased* dataset, but other versions exist, including QMNIST (Yadav and Bottou, 2019), where the authors reconstructed the MNIST test set with 60,000 samples.

Another successful challenge series is the KDD Cup, with its first edition taking place in 1997[10]. KDD Cup has focused on challenges on data mining bridging industry and academy, with a variety of topics being covered with time, from retailing, recommendation and customer analysis to authorship analysis and student performance evaluation[11]. While KDD Cup has been more application-oriented, findings from this competition have resulted in progress in the field without any doubt. KDD Cups are reviewed in the next chapter.

The first decade of the 2000 was critical for the consolidation of challenges as a way to solve tough problems with the help from the community. It was during this time that the popular Netflix prize[12] was organized, granting a 1M dollar prize to the team able to improve the performance of their *in-house* recommendation method. The winning team improved by $\approx 10\%$ the reference model (Koren, 2009). Also, one of the long-lived competition programs in the context of machine learning arose in this decade[13]: the *ECML/PKDD Discovery Challenge series*. Organized since 1999, this forum has released a number of

---

8. https://semeval.github.io/
9. http://yann.lecun.com/exdb/mnist/
10. https://www.kdd.org/kdd-cup/view/kdd-cup-1997
11. https://kdd.org/kdd-cup
12. https://www.netflixprize.com/
13. https://sorry.vse.cz/~berka/challenge/PAST/

datasets, although it is now an established competition track, in the early years, competitions consisted of releasing data and asking participants to build and evaluate solutions by themselves. The NeurIPS 2003 feature selection challenge took place[14] in this decade too, being this one of the oldest machine learning competitions in which test data was withheld from participants (Guyon et al., 2004).

In that same decade, the first edition of evaluation efforts that are still being run were launched, for instance, the first: CLEF[15] (2000), ImageCLEF[16] forum (2003), TRECVID[17] conference (2003), PASCAL VOC[18] (2005) challenges. All of these efforts and others that evolved over the years (e.g., the model selection[19] and performance prediction[20] challenges (2006) that laid the foundation for AutoML challenges), set the basis for the settlement of academic competitions.

The 2000s not only were fruitful in terms of the number and variety of long lasting challenges that emerged, but also because of the establishment of organizations. It was in 2009 that Kaggle[21] was founded, initially focused on challenges as a service, nowadays Kaggle also offers training-learning, hiring and data-code sharing options. From the academic side, in 2011 ChaLearn[22], the *Challenges in Machine Learning Organization* was founded as well. ChaLearn is a non-profit organization that focuses on the organization and dissemination of academic challenges. ChaLearn provides support to potential organizers of competitions and regularly collaborates with a number of institutions and research groups, likewise, it focuses on research associated to challenge organization in general, this book is a product of such efforts.

From 2010 and on challenges have been established as one of the most effective way of boosting research in a specific problem to get practical solutions rapidly. The ImageNet Large Scale Visual Recognition Challenge (ILSVRC) featured from 2010 to 2017 has been among the most successful challenges in computer vision, as it witnessed the rise of CNNs for solving image classification tasks, see next section. Likewise, the VOC challenge organized until 2012, contributed to the development of object detection techniques like YOLO (Redmon et al., 2016). The AutoML challenge series (from 2015) proved that long term contests with code submission could lead to progress on the automation of model design at different levels. As a result, nowadays, top-conferences and venues from different fields have their competition track. Table 1 shows representative competition programs associated to major conferences and related organizations.

This table illustrates that many scientific communities have acknowledged the importance of academic competitions, and highly value these by dedicating resources towards organizing such competitions. Please note that there are top tier venues that do not have an *official* competition track, and therefore they were not included in this table. However,

---

14. `http://clopinet.com/isabelle/Projects/NIPS2003/`
15. `https://www.clef-initiative.eu/web/clef-initiative/`
16. `https://www.imageclef.org/`
17. `https://trecvid.nist.gov/`
18. `http://host.robots.ox.ac.uk/pascal/VOC/`
19. `http://clopinet.com/isabelle/Projects/NIPS2006/home.html`
20. `http://www.modelselect.inf.ethz.ch/`
21. `https://kaggle.com/`
22. `http://chalearn.org/`

Table 1: Competition tracks of main conferences in machine learning and related fields. Column four shows the number of tasks organized in the latest edition of the associated track (# Tasks LE) as of 2025 or 2026. Acronyms are as follows: Machine Learning (ML), Data Mining (DM), Computational Intelligence (CI), Pattern Recognition (PR), Robotics (RO), MIR (Multimedia Information Retrieval), Multimedia Information Processing (MIP), Information Retrieval (IR), Natural Language Processing (NLP), Artificial Intelligence (AI), Evolutionary Computation (EC), Medical Image Analysis (MI), Signal Processing (SP), Image Processing (IP), Miscellaneous (MS). The last four rows of this table shows institutions and organizations associated with challenges.

| Venue | Field | Since | # Tasks LE | URL |
|---|---|---|---|---|
| TREC | IR | 1993 | 7 | https://trec.nist.gov/ |
| ICDAR | PR | 1993 | 8 | https://icdar2026.org/ |
| KDD | DM | 1997 | 2 | https://kdd.org/ |
| ECML | ML | 1999 | 3 | https://ecmlpkdd.org/ |
| RoboCup | RO | 1997 | 5 | https://www.robocup.org/ |
| PAN-CLEF† | NLP | 2000 | 5 | https://pan.webis.de/ |
| TrecVid | MIR | 2003 | 2 | https://trecvid.nist.gov/ |
| ImageCLEF† | MIP | 2003 | 5 | https://www.imageclef.org |
| MediaEval | MIP | 2003 | 6 | https://multimediaeval.github.io/ |
| GECCO | EC | 2004 | 12 | https://gecco-2026.sigevo.org/HomePage |
| WCCI | CI | 2006 | 4 | https://attend.ieee.org/wcci-2026/ |
| MICCAI | MI | 2007 | 38 | https://conferences.miccai.org/2022/en/ |
| Interspeech | SP | 2008 | 7 | https://interspeech2026.org/en-AU |
| ICRA | RO | 2008 | 9 | https://2026.ieee-icra.org/ |
| ACM Multimedia | MIP | 2009 | 25 | https://2026.acmmm.org/ |
| ICPR | PR | 2010 | 5 | https://icpr2026.org/ |
| SemEval | NLP | 2010 | 13 | https://semeval.github.io/ |
| IROS | RO | 2012 | 8 | https://2026.ieee-iros.org/ |
| ICMI | MIP | 2013 | 3 | https://icmi.acm.org/2026/ |
| ICASSP | SP | 2014 | 14 | https://2026.ieeeicassp.org/ |
| ICME | MIP | 2015 | 8 | https://2026.ieeeicme.org/ |
| CIKM | DM | 2017 | 1 | https://cikm2026.diag.uniroma1.it/ |
| ICIP | IP | 2017 | 4 | https://2026.ieeeicip.org/ |
| NeurIPS | ML | 2018 | 18 | https://neurips.cc/Conferences |
| IJCAI | AI | 2018 | 14 | https://2026.ijcai.org/ |
| AutoML | ML | 2022 | 1 | https://automl.cc/ |
| Loingitude Prize | MS | 1714* | 1 | https://longitudeprize.org/ |
| XPrize* | MS | 1996 | 2 | https://www.xprize.org/ |
| Kaggle | MS | 2009 | - | https://www.kaggle.com/ |
| ChaLearn | ML | 2011 | - | http://chalearn.org/ |

these venues have hosted workshops associated to competitions that have had great impact. Just to name a few: CVPR, ICCV, ECCV, ICML, ICLR, EMNLP, ACL.

## 2.2 Progress driven by academic challenges

As previously mentioned, challenges are now established mechanisms for dealing with complex problems in science and industry. This is not fortuitous, but a response from the community to a number of accomplishments in different fields. This section aims to briefly summarize the main achievements of selected challenges that have motivated other researchers and fields to organize competitions. We focused on a representative machine learning challenge (AutoML) and two evaluation campaigns from the two fields where more contests are organized, see Figure 2.

- **AutoML challenges.** AutoML is the sub field of machine learning that aims at automating as much as possible all of the aspects of the design cycle (Hutter et al., 2018). While people were initially skeptical of the potential of this sort of methods, nowadays AutoML is a trending research topic within machine learning (there is a dedicated AutoML conference with a competition track[23] since 2022). This is in large part due to the achievements obtained in the context of AutoML challenges. Back in 2006 early efforts in this direction were the prediction performance challenge (Guyon et al., 2006) and the agnostic *vs.* prior knowledge challenge (Guyon et al., 2008). These contests asked participants to build methods for automatically or manually building classification models. They became the predecessors of the AutoML challenge series that ran from 2015 to 2018 (Guyon et al., 2019), and all of the follow up events that are still organized. Initially, the AutoML challenge series focused on tabular data, but it then evolved to deal with raw heterogeneous data in the AutoDL[24] challenge series(Liu et al., 2021b), whose latest edition is the Cross-Domain MetaDL challenge 2022 [25] (El Baz et al., 2021a,b; Carrión-Ojeda et al., 2022). A number of methods (e.g., AutoSKLearn (Feurer et al., 2019)), evaluation protocols, AutoML mechanisms (e.g., Fast Augmentation Learning methods (Baek et al., 2020)) and improvements arose in the context of these challenges including the evaluation of submitted code, cheating prevention mechanisms, the progressive automation of different types of tasks (e.g., from binary classification to regression, to multiclass classification, to neural architecture search) and the use of different data sources (from tabular data, to raw images, to raw heterogeneous datasets). The result is an established benchmark that is widely used by the community.

- **ImageNet Large Scale Visual Recognition Challenge.** The so called, ImageNet challenge asked participants to develop image classification systems for 1,000 categories and using millions of images as training data (Russakovsky et al., 2015). At the time of the first edition of the challenge, object recognition, image retrieval and classification datasets were dealing with problems involving thousands of images and dozens of categories (see e.g., (Escalante et al., 2010)). While the scale made participants struggle in the first two editions of the challenge, the third round witnessed the renaissance of convolutional neural networks, when AlexNet reduced drastically the error rate for this dataset (Krizhevsky et al., 2012). In the following editions of the challenge other landmark CNN-based architectures for image classification were proposed including: VGG (Simonyan and Zisserman, 2015), GoogLeNet (Szegedy et al., 2015) and ResNet (He et al., 2015). These architectures comprised important contributions to deep learning, including residual connections/blocks and inception-based networks, the establishment of regularization mechanisms like dropout, pretraining and fine tuning and the efficient usage of GPUs for training large models. While the challenge itself did not provoke the aforementioned contributions, it was the catalyst and solid test bed for the rise of deep learning in computer vision.

---

23. `https://automl.cc/`
24. `https://autodl.chalearn.org/`
25. `https://metalearning.chalearn.org/`

- **Text Retrieval Evaluation Conference.** TREC initially focused on the evaluation of information retrieval systems (text) (see (Voorhees and Harman, 2005; Rowe et al., 2010) for an overview of the early editions of TREC), but it rapidly evolved to include novel tasks and evaluation scenarios in the forthcoming years. This led to include? tasks that involved information sources from multiple languages, and eventually images and videos. Other tasks that have been widely considered in the TREC campaign are: question answering, adaptive filtering, text summarization, indexing, among many others. Thanks to this effort the information retrieval and text mining fields were consolidated and boosted the progress in the development of search engines and related tools that are quite common nowadays. Well known retrieval models and related mechanisms for efficient indexing, query expansion, relevance feedback, arose in the context of TREC or were validated in this forum. Another important contribution of TREC through the years is that it has evolved to give rise to numerous tasks and application scenarios that have defined the text mining field.

We surveyed a few representative challenges and outlined the main benefits that they bring into their respective communities. While these are very specific examples and while we have chosen breaking through competitions, similar outcomes can be drawn from challenges organized in other fields. In Section 3 we review challenges from a wider variety of domains.

## 2.3 Pros and cons of academic challenges

We have learned so far that challenges are beneficial in a number of ways, and have boosted progress in a variety of domains. However, it is true that there are some limitations and undesired effects of challenges that deserve to be pointed out. This section briefly elaborates on benefits and limitations of academic challenges.

### 2.3.1 Benefits of academic challenges

The main benefit of challenges is the solution of complex problems via crowd sourcing, advancing the state of the art and the establishment of benchmarks. There are, however, other benefits that make them appealing to both participants and organizers, these include:

- **Training and learning through challenges.** Competitions are an effective way to learn new skills, they *challenge* participants to gain new knowledge and put in practice known concepts for solving relevant problems in research and industry. Even if participants do not win a challenge or a series of them, they progressively improve their problem solving skills.

- **Challenges are open to anyone.** Apart of political restrictions that may be applied for some organizations, competitions target anyone with the ability to approach the posted problem. This is particularly appealing to underrepresented groups and people with limitations to access the cutting edge problems, data and resources. For instance, most competitions adopting code submission provide cloud-based computing to participants. Likewise, challenges can be turned into ever lasting benchmarks and they contribute to making data available to the public.

- **Engagement and motivation.** The engagement offered by competitions is priceless. Whether the reward is economic, academic (e.g., publication or talk in a workshop, professional recognition in the field), competitiveness, or just fun, participants find challenges motivating.

- **Reproducibility.** This cannot be emphasized enough, benchmarks associated to challenges not only provide the task, data and evaluation protocols. In most cases resources, starting-kits, others' participants code and computing resources are given as well. This represents an easy way to get into competitions to participants, which can directly compete with state-of-the-art solutions. At the same time, competitions having these features guarantee reproducibility of results which is clearly beneficial to the progress in the field.

### 2.3.2 PITFALLS OF ACADEMIC CHALLENGES

Despite the benefits of challenges, they are not risk-free, therefore, there are certain limitations that should be taken into account.

- **Performance improvement vs. scientific contribution.** Academic challenges often ask participants to build solutions that achieve the best performance according to a given metric. Although in most cases there is a research question associated to a challenge, participants may end up building solutions that optimize the metric but that do not necessarily result in new knowledge. This gives challenges a bitter-sweet taste, as often new findings are overshadow by super-tuned off-the-shell solutions.

- **Stagnation.** An undesirable outcome for a challenge is stagnation, this is often the result of wrong challenge design decisions, that result in either a problem that is too hard to be solved with current technology or unattractive to participants. While it is not possible to anticipate how far the community can go in solving a task, the implementation of (strong) baselines, starting kits and appealing datasets, or rewards could help to avoid stagnation.

- **Data Leakage.** It refers to the use of target (or any other relevant information that is supposed to be withheld from participants) information by participants to build their solutions (Kaufman et al., 2011). This is a common issue when datasets are re-used or when datasets are build from external information (e.g., from social networks). Anonymization and other mechanisms as those exposed in (Kaufman et al., 2011) could be adopted for avoiding this problem.

- **Privacy and rights on data.** *"Data is the new oil"* has been a popular say recently[26], while this is debatable, it is true that data is a valuable asset that must be *handled with care*. Therefore copyright infringement should be avoided to the uttermost end. Likewise, failing to guarantee privacy is an important issue that must be addressed by organizers as this could lead into legal issues. Anonymization mechanism should be applied to data before its release, making sure it is not possible to track users identity or other important and confidential information.

---

26. https://www.forbes.com/sites/forbestechcouncil/2019/11/15/data-is-the-new-oil-and-thats-a-good-thing/?sh=381ec30d7304

**Additional Downsides from the Perspective of Participants** We now list additional drawbacks from the participants' perspective that should be taken into account when designing and running an academic competition.

- **The cost-benefit asymmetry of participation.** Participating in a competition imposes substantial costs on participants. However, only few participants are *rewarded* or *recognized*.

- **The compute gap.** Some competitions do not provide cloud-computing resources for participants. In this scenarios there is a clear asymmetry between well-resourced and under resourced participants.

- **Benchmark contamination and data leakage in the LLM era.** The widespread practice of relying on pretrained models introduces the risk that some test-set documents or samples were considered as pretraining data.

- **Gaming the leaderboard.** Overfitting of the learboard can occur on competitions implementing public leaderboards and unlimited submissions.

- **Psychological and well-being concerns.** Competitive environments can generate significant psychological pressure, particularly for participants who have invested heavily in a challenge.

- **Intellectual property and credit attribution.** When participants submit code or models as part of a competition, the intellectual property (IP) arrangement is not always clearly defined.

- **The saturation problem from a participant perspective.** Once a benchmark is saturated (i.e., top-performing systems approach or exceed human-level performance) the marginal scientific value of further improvement is low.

These shortcomings suggest that a more participant-centered approach to competition design (attending to compute equity, reproducibility, psychological well-being, and clear IP frameworks) would help to ensure that the benefits of academic challenges are distributed more equitably and that the participation experience reinforces, rather than undermines, participants' long-term engagement with research.

## 2.4 What makes academic challenges successful?

Having reviewed competitions, their benefits and pitfalls/limitations, this section elaborates on characteristics that we think make a challenge successful. While it is subjective to define a successful challenge, the following guidelines associate success to high participation, quantitative performance and novelty of top ranked solutions.

- **Scientific rigor.** The design and the analysis of the outcomes of a competition are critical for its success. Following scientific rigor as "to ensure robust and unbiased experimental design, methodology, analysis, interpretation and reporting of results" (Hofseth, 2017) is necessary and helps to avoid some of the limitations mentioned above. Adopting statistical testing for the analysis of results, careful designing

of evaluation metrics, establishing theoretical bounds on these, running multiple tests before releasing the data/competition, formalizing the problem formulation, performing ablation studies are all critical actions that impact on the outcomes of academic challenges.

- **Rewarding and praising scientific merit and novelty of solutions.** It is worth mentioning that novel methods do not always make it to the top of the leaderboard, but these new ideas may be great seeds and serve as an inspiration to others for further fruitful research. Therefore, rewarding and acknowledging scientific merit and novelty of solutions is very important. There are several ways of doing this, for instance, having a *prize* for the most original/novel submission or granting a best paper award that is not entirely based on quantitative performance.

- **Publication and dissemination of results** are good practices with multiple benefits. Participants are often invited to fill out *fact sheets* and write workshop papers in order to document their solutions. Similarly, organizers commonly publish overview papers that summarize the competition, highlighting the main findings and analyzing results in detail. Associating a special issue of a journal with competitions is a good idea as it is motivating for participants, and at the same time it is a *product* that organizers can report in their work evaluations.

- **Associating the competition with an top tier venue** (e.g., conferences, summits, workshops, etc.) makes a challenge more attractive to participants, as they associate the quality of associated venues and competitions. Also, physically attending the competition session is more appealing if participants can also attend top tier events.

- **Organization of panels and informal discussion sessions** involving both participants and organizers is valuable for sensing perception of people associated to the event. This is critical when organizing challenges that run for several editions.

- **Establishing benchmarks** should be an underlying goal of every competition. Therefore, curated data, fail safe evaluation protocols, and adequate platforms for maintaining competitions as long term evaluation test beds are essential. Likewise, the use of open data and open source code for the purposes of reproducibility and so that everyone can benefit and continue their own research.

### 2.4.1 ACADEMIC VS. INDUSTRIAL CHALLENGES

Industrial challenges are described in detail in the next chapter. In this section we outline the main differences of industry and scientific competitions.

The main objective of industrial challenges is the economic advantage from the winning model that will potentially increase profits and improve business model, meaning it should be an end-to-end solution. The organizers care much less about scientific publications, being scientifically rigorous neither about the results being statistically significant. These types of contest do not single out scientific questions, that is not the priority for them. They aim at specific business problems, usually posses big not preprocessed datasets and evidently can provide more often big prizes. Up till now the direct positive correlation

between these big rewards and qualitative contributions has not been proven. But it was observed that big prizes might attract many participants, create "big splash" in the news for the company-organizer and cause a lot of noise in the leaderboard, potentially leading to gaining by chance. While the winners and contributors of academic challenges get scientific recognition, the top performers at the industrial contests can receive job offers and be hired by the organizers.

Another important aspect of industrial challenges is that due to their nature and the concurrent market, the company-organizers prefer to keep the data and the submitted code private, which is in the opposition with the scientific mentality, because it prevents to benefit from the latest break-through and get inspiration from the newest ideas.

## 3 Academic challenges across different fields

This section briefly reviews challenges across different fields. We focus on fields that have long tradition in challenges. In order to identify such fields of knowledge, we surveyed competitions organized in the CodaLab platform (Pavao et al., 2023). Figure 2 shows a distribution of CodaLab challenges across fields of knowledge. Clearly NLP and Computer vision challenges dominate, this could be due to the explosion of availability of visual and textual data of the last few years. One should note that most of the competitions shown in that plot have a strong machine learning component. In the remainder of this section we briefly survey competitions organized in a subset of selected fields.

### 3.1 Challenges in Machine Learning

Machine learning is a transversal field of knowledge that has been present in most challenges regardless of the application field (e.g., computer vision, OCR, NLP, time series analysis, and so on). Therefore, it is not easy to cast a challenge as a ML competition. For that reason, in this section we review as a representative sample the competition track of the NeurIPS conference. The track has run regularly since 2017, although challenges organized with the conference date back to the early 2000s (Guyon et al., 2004). Overview papers for the NeurIPS competition track from 2019 to 2023 can be found in (Escalante and Hadsell, 2019; Escalante and Hofmann, 2020; Kiela et al., 2022; Ciccone et al., 2023).

Figure 3 shows the number of competitions that have been part of the NeurIPS competition track. There has been an increasing number of competitions organized each year, see also (Carlens, 2023) for more details. The topics of challenges are quite diverse, with deep reinforcement learning (DRL) prevailing since the very beginning of the track. The first competition in the program around this topic was the Learning to Run challenge[27] that asked participants to build an human-like agent to navigate an environment with obstacles (Kidzinski et al., 2018), this challenge was run for two more editions, the last one being the Learn to Move - Walk Around[28] challenge. DRL-based competitions addressing other challenging navigation scenarios are the Animal Olympics[29] and MineRL series, see below. DRL challenges addressing different tasks are the Real robot challenge[30] series with

---

27. `https://www.aicrowd.com/challenges/nips-2017-learning-to-run`
28. `https://www.aicrowd.com/challenges/neurips-2019-learn-to-move-walk-around`
29. http://animalaiolympics.com/AAI/
30. `https://real-robot-challenge.com/`

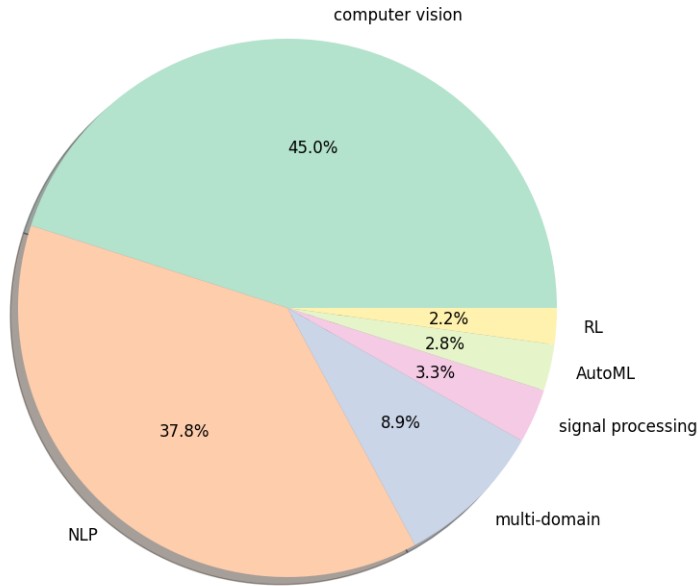

Figure 2: Distribution of competitions with different machine learning domains. Data gathered from CodaLab Competitions (Pavao et al., 2023)

two editions, the Learning to run a power network competition[31] and the two editions of the Pommerman[32] competition where the goal was to develop agents to compete to each other in a bomberman-game-like scenario. The presence of DRL in the challenge track as been growing in the last editions.

Another popular topic in the NeurIPS competition track is AutoML: since 2018, at least one competition associated to this topic has been part of the NeurIPS competition track. These include the AutoML@NeurIPS (Escalante et al., 2019) and AutoDL (Liu et al., 2021b) challenges, the black-box optimization competition (Turner et al., 2021), the predicting generalization in deep learning challenge [33], two editions of the Meta-DL challenge (El Baz et al., 2021b; Carrión-Ojeda et al., 2022) and the AutoML Decathlon[34].

Specific challenges that have been part of the competition track for more than 2 editions are the following:

- **Traffic4cast**[35]. Organizing variants of challenges aiming to predict traffic conditions under different settings and scenarios, see (Kreil et al., 2020; Kopp et al., 2021; Eichenberger et al., 2022).

---

31. https://l2rpn.chalearn.org/

32. https://www.pommerman.com/

33. https://sites.google.com/view/pgdl2020

34. https://www.cs.cmu.edu/~automl-decathlon-22/

35. https://www.iarai.ac.at/traffic4cast/

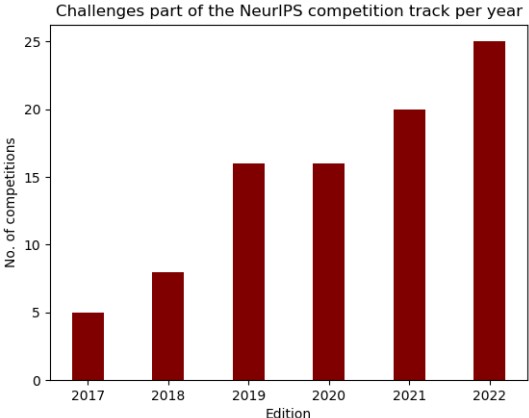

Figure 3: Number of challenges organized as part of the NeurIPS competition program.

- **The AI Driving Olympics (AI-DO).** Aiming to build autonomous driving systems running in simulation and small physical vehicles tested live during the competition track[36].

- **MineRL**[37] A competition series focusing on building autonomous agents that using minimal resources are able to solve very complex tasks in a MineCraft environment. In the first two editions agents were asked to find a diamond with limited resources, see (Milani et al., 2020; Guss et al., 2021). In the most recent editions tasks have been varied and more specific (Shah et al., 2022).

- **Reconnaissance Blind Chess.** Challenges participants to build agents able to play a chess variant in which a player cannot see her opponent's pieces but can learn about them through private, explicit sensing actions. Three editions of this competition have run in the track (Gardner et al., 2020).

It is difficult to summarize the number and variety of topics addressed in challenges part of the NeurIPS competition, however, we have reviewed a representative sample. Nevertheless, please note that most challenges reviewed in the remainder of this section also include an ML component.

### 3.2 Challenges in Computer Vision

Together with machine learning, computer vision has been greatly benefited from challenges. As previously mentioned, The PASCAL Object detection challenge series boosted research on object detection and semantic segmentation (Everingham et al., 2015). The ImageNet large scale classification challenge is another landmark competition that served as platform for the renaissance of convolutional neural networks (Russakovsky et al., 2015). In addition to these landmark competitions there have been a number of efforts that have pushed further the state-of-the-art, these are reviewed in the following lines.

---

36. https://www.duckietown.org/research/AI-Driving-olympics
37. https://minerl.io/

The ChaLearn Looking at People (ChaLearn LAP[38]) series has organized academic challenges around the analysis of human behavior from visual information. More than 20 competitions on the topic have been organized so far, see (Escalera et al., 2017a) for a (outdated) review. Among the organized competitions several of the datasets have become a reference for different tasks, and are used as benchmarks. These include: the gesture recognition challenges (Escalera et al., 2013, 2014, 2017b; Wan et al., 2017), the personality recognition challenge series (Escalante et al., 2017, 2022; Palmero et al., 2021), the age estimation challenge series (Escalera et al., 2015, 2016) and the face anti-spoofing challenge series (Liu et al., 2019; Wan et al., 2020; Liu et al., 2021a). A wide diversity of related topics have been studied in the context of ChaLearn LAP challenges, including: action recognition and cultural event recognition (Baró et al., 2015; Escalera et al., 2015), sign language understanding (Sincan et al., 2021), identity preserving human analysis (Clapés et al., 2020) among others. Undoubtedly, these challenges have advanced the state of the art in a number of directions within computer vision and affective computing.

The Common Objects in COntext (COCO[39]) challenge series that emerged after the end of the Pascal VOC challenge. This effort continued benchmarking object detection methods, but also started evaluating the so called *image captioning* task. Early efforts for the evaluation of this task emerged in the ImageCLEF forum (Clough et al., 2010; Escalante et al., 2010), where the goal was associating keywords to images. The COCO challenge was more ambitious by asking participants to describe the content of an image with a more *human-like* description. Running from 2015-2020 this benchmark was critical for the consolidation of the image captioning task, with major contributions being reported at the beginning of the series, see (Bai and An, 2018; Stefanini et al., 2021). Today, COCO is an established benchmark in a number of tasks related to vision and language, see (Lin et al., 2014).

Other efforts in the field of computer vision are the NTIRE challenge, focused on image restoration, super resolution and enhancement (Timofte et al., 2017; Gu et al., 2022) , the visual question answering competition[40] running from 2016 to 2021, the fine grained classification workshop [41] that has run a competition program since 2017, the EmotioNet[42] recognition challenge that ran in 2020 and is now a testbed for emotion recognition, the ActivityNet challenge [43] organized since 2016 and targeting action recognition in video, among several others.

## 3.3 Challenges in Natural Language Processing

The development of the natural language processing (NLP) field, in particular for text mining and related tasks, has been largely driven by competitions, also known in the NLP jargon as *shared tasks*. In fact, one of the oldest evaluation forums across all computer science is one focusing in NLP, that is TREC. It initially focused on the evaluation of information retrieval systems (text), but it rapidly evolved to include novel tasks and evaluation scenar-

---

38. https://chalearnlap.cvc.uab.cat/
39. https://cocodataset.org
40. https://visualqa.org/
41. https://sites.google.com/view/fgvc9
42. https://cbcsl.ece.ohio-state.edu/enc-2020/index.html
43. http://activity-net.org/challenges/2022/

ios in the forthcoming years (Voorhees and Harman, 1998, 2005; Rowe et al., 2010). This lead to consider tasks that involved information sources from multiple languages (Harman, 1998), and eventually, speech signals (Garofolo et al., 2000) and visual information (Awad et al., 2021). Other tasks that have been considered in the TREC campaign are: question answering (Voorhees, 2001), adaptive filtering (Harman, 1995), text summarization [44], among many others. Thanks to this effort the information retrieval and text mining fields were consolidated and boosted the progress in the development of search engines and related tools that are quite common nowadays.

Several well known evaluation campaigns evolved from TREC and consolidated on their own. Most notably, the TRECVid (Awad et al., 2021) and Cross-Language Evaluation Forum (Braschler, 2001) (CLEF) campaigns. The former focusing on tasks related to video retrieval, indexing and analysis. The academic and economic impact of TRECVid has been summarized already. Showing the relevance that such forum has had into the progress of video search technology. CLEF is another forum that initially focused on cross-lingual text analysis tasks. Now it is a conference that comprises several shared tasks, called labs. This include ImageCLEF, PAN among others. Likewise, there are forums dedicated to specific languages, for example, Evalita[45] (for Italian), IberLEF[46] (for Spanish) and GermEval[47].

In terms of speech, there were several efforts from DARPA (Marcus, 1992; Black and Eskenazi, 2009) and NIST[48] in organizing competitions as early as the late 80s. These long term efforts have helped to shape ASR and related fields. More recently, after the deep learning empowering, several challenges focusing on speech have been proposed, these are often associated to major conferences in the field (e.g. Interspeech and ICASSP), see Table 1. There is no doubt that competitions have played a key role for the shaping the wide field of NLP.

## 3.4 Challenges in Biology

Biology is a field of knowledge that has benefited from competitions considerably. In terms of medical imaging, the premier forum is the grand challenge series associated to the MICCAI conference, running since 2007 [49]. A number of important challenges have been organized in this context, where most competitions deal with medical imagery segmentation or reconstruction of different organs, body parts and input type, see e.g., (Scully et al., 2008; Marak et al., 2009; Andrearczyk et al., 2022). In recent editions the challenge scenarios and approached tasks have been increasing difficulty and the potential impact of solutions. In its last edition, the MICCAI grand challenge series has 38 competitions running in parallel. This is an indicator of success among the medical imaging community.

Other challenges associated to medical image analysis have been presented in forums associated to image processing and computer vision as well. For instance, in 2019 during The IEEE International Symposium on Biomedical Imaging (ISBI), nine challenges were

---

44. http://trecrts.github.io/
45. https://www.evalita.it/campaigns/evalita-2022/
46. https://sites.google.com/view/iberlef2022
47. https://germeval.github.io/
48. https://www.nist.gov/itl/iad/mig/past-hlt-evaluation-projects
49. https://cause07.grand-challenge.org/Results/

organized[50]. In 2020 a challenge on Image processing on real-time distortion classification in laparoscopic videos was organized with ICIP 2020[51]. In the context of ICCV, challenges on remote measurement of physiological signals from videos (RePSS) were organized: one on measurement of inter-beat-intervals (IBI) from facial videos, and another one on respiration measurement from facial videos (Li et al., 2020, 2021).

It is worth mentioning that there are platforms associated with challenges in biology and medical sciences. The Grand Challenge[52] platform being perhaps the oldest one and the most representative in terms of imagery: *more than 150 competitions are listed in the platform*, most of which are associated to medical image analysis. A related effort is that of the DREAM challenges[53] a platform that has organized more than 60 challenges in biology and medicine. The variety of topics covered by DREAM challenges is vast (Stolovitzky et al., 2009): from systems biology modelling (Meyer and Saez-Rodriguez, 2021), to prevention (Tarca et al., 2020) and monitoring (Sun et al., 2022) damage caused by certain conditions, to disease susceptibility[54], to analyzing medical documents with NLP[55], to drug analysis and combination[56] and many other relevant topics. As seen in Chapter 5, platforms play a key role in challenge success, biology is a field where excellent platforms are available and this has been critical for the advancement of state of the art in this relevant field.

Protein structure modelling was officially introduced in 1994 at the biennial large-scale experiment Critical Assessment of protein Structure Prediction (CASP), and ever since it attracted more than 100 teams to tackle the problem, see (J et al., 2014). Only almost 20 years later, two teams presented breaking through solutions to protein folding task (Kryshtafovych et al., 2021): DeepMind with their AlphaFold2 (Jumper et al., 2021) and scientists of the University of Washington with RoseTTAFold (Baek et al., 2021). Alphafold uses multiple neural networks that feed into each other in two stages. It starts with a network that reads and folds the amino acid sequence and adjusts how far apart pairs of amino acids are in the overall structure. Then goes the structure model network that reads the produced data, creates a 3D structure, and makes the needed adjustments (Evans et al., 2018; Jumper et al., 2021). RoseTTAFold adds a simultaneous third neural network, which tracks where the amino acids are in 3D space as the structure folds, alongside the 1D and 2D information (Baek et al., 2021). The solution of Washington University is less accurate but uses less computational and time resources than AlphaFold2. Without the existence of the CASP experiment, achieving the outstanding performance of these methods would have taken much more time.

As we can see, advancements of machine learning in biology are of crucial importance, that's why there are numerous competitions in this domain. Researchers and practitioners are trying to deal with biological and related domain (medicine, agriculture, and others) challenges using various machine learning solutions like computer vision, NLP and signal processing.

---

50. `https://biomedicalimaging.org/2019/challenges/`
51. `https://2020.ieeeicip.org/challenge/real-time-distortion-classification-in-laparoscopic-videos/`
52. `https://grand-challenge.org/challenges/`
53. `https://dreamchallenges.org/closed-challenges/`
54. `https://dreamchallenges.org/respiratory-viral-dream-challenge/`
55. `https://dreamchallenges.org/electronic-medical-record-nlp-dream-challenge/`
56. `https://dreamchallenges.org/astrazeneca-sanger-drug-combination-prediction-dream-challenge/`

### 3.5 Challenges in Autonomous Driving

DARPA Grand Challenge is considered as one of the first long distance race for autonomous driving cars, it was organised in 2004 with more than 100 teams. None of the robot vehicles managed to finish the 240 km route, only one member covered 11.78 km and then got stuck. Next year there were 195 teams, the distance of the challenge was of 212 km, and five vehicles successfully completed the course. These first courses were challenging but vehicles "operated in isolation", their interaction was not required, and there was no traffic either. So the next Urban challenge was held in 2007 in a city area, the objective was to complete 96km in 6 hours and it included "driving on roads, handling intersections and maneuvering in zones" (Urmson et al., 2007). Six teams managed to complete the course.

The basics were laid, and DARPA pursued their competitions: Robotics Challenge in 2012, 2013 - Fast Adaptable Next-Generation Ground Vehicle Challenge, 2013 – 2017 Subterranean Challenge on "autonomous systems to map, navigate, and search underground tunnel, urban, and cave spaces" [57].

Being able to test autonomous driving cars "in the wild" is important and expensive. In order to fine-grain the algorithms at a less cost one needs to test them virtually. Hopefully, there are different simulators: CARLA [58], VISTA 2.0 [59], NVIDIA DRIVE Sim [60] and others.

Several challenges have been organised based on CARLA simulator, "an open-source simulator for autonomous driving research", which is used to study "a classic modular pipeline, a deep network trained end-to-end via imitation learning, and a deep network trained via reinforcement learning" (Dosovitskiy et al., 2017).

Autonomous driving has numerous interesting challenges, and object detection is one of them. Most of the current research concentrates around camera images, but it is not the best sensor under certain conditions like bad weather, poor lighting. Radar information can help to overcome these inconveniences. It is more reliable, cost-efficient and might potentially lead to better object detection. ROD2021 Challenge is the first competition of its' kind, which proposes object detection task on radar data, and was held in the ACM International Conference on Multimedia Retrieval (ICMR) 2021. Organisers developed their own baseline: "radar object detection pipeline, which consists of two parts: a teacher and a student. Teacher's pipeline fuses the results from both RGB and RF images to obtain the object classes and locations in RF images. Student's pipeline utilizes only RF images as the input to predict the corresponding ConfMaps under the teacher's supervision. The LNMS as post-processing is followed to calculate the final radar object detection results." (Wang et al., 2021b).

This challenge attracted more than 260 participants among 37 teams with around 700 submissions. The winning team, affiliated to Baidu, submitted paper "DANet: Dimension Apart Network for Radar Object Detection" (Ju et al., 2021), where they presented their results. "This paper proposes a dimension apart network (DANet), including a lightweight dimension apart module (DAM) for temporal-spatial feature extraction. The proposed DAM extracts features from each dimension separately and then concatenates the features

---

57. https://www.darpa.mil/about-us/subterranean-challenge-final-event

58. https://carla.org/

59. https://vista.csail.mit.edu

60. https://www.nvidia.com/en-us/self-driving-cars/simulation/

together. This module has much smaller number of parameters, compared with RODNet-HGwI, so that significant reduction of the computational cost can be achieved. Besides, a vast amount of data augmentations are used for the network training, e.g., mirror, resize, random combination, Gaussian noise and reverse temporal sequence. Finally, an ensemble technique is implemented with a scene classification for a more robust model. The DANet achieves the first place in the ROD2021 Challenge. This method has relatively high performance but with less computational cost, which is an impressive network model. Besides, this method shows data augmentation and ensemble techniques can greatly boost the performance of the radar object detection results" (Wang et al., 2021a).

Another interesting and pioneering challenge is OmniCV (Omnidirectional Computer Vision) in conjunction with IEEE Computer Society Conference on Computer Vision and Pattern Recognition (CVPR'2021). The objective was to evaluate semantic segmentation techniques targeted for fisheye camera perception. It attracted 71 teams and a total of 395 submissions. Organisers proposed their baseline "a PSPNet network with a ResNet50 backbone finetuned on WoodScape Dataset", which "achieved a score of 0.56 (mIoU 0.50, accuracy 0.67) excluding void class". The top teams managed to get significantly better scores and proposed interesting solutions. The winning team implemented full Swin-transformer Encoder-Decoder approach, with a score of 0.84 (mIoU 0.86, accuracy 0.89) (Ramachandran et al., 2021).

## 4 Academic Competitions in the Era of Foundation Models

The preceding sections have focused on academic competitions from their early years through their consolidation as a mainstream mechanism for advancing machine learning and related fields. The landscape is, however, far from static. Three developments in particular are reshaping the nature, scope, and methodology of academic challenges in ways that warrant dedicated attention. This section elaborates on these critical aspects that represent the frontier of academic competitions.

### 4.1 Challenges in the Large Language Models era

The rise and establishment of large language models (LLMs) has generated an entirely new family of academic challenges, while simultaneously reshaping the evaluation landscape of existing ones. Prior to the availability of effective instruction-following models, NLP competitions largely evaluated task-specific fine-tuned systems on well-defined benchmarks (see Section 3.3). With the emergence of models such as GPT, Claude, Gemini, LLaMA, among many others, the community has had to rethink what evaluation means, how leaderboards can be kept meaningful, and what kinds of problems are worth posing to the crowd. In the remainder of this section we elaborate on relevant aspects for competitions in the context of LLMs.

#### 4.1.1 Benchmark-style LLM evaluations

Some of the most influential LLM "competitions" are not hosted on a challenge platform in the traditional sense, but rather take the form of *open leaderboards* where any team can submit model outputs for evaluation against a held-out test set. The General Language Un-

derstanding Evaluation (GLUE) benchmark and its successor SuperGLUE were pioneers on this format (Wang et al., 2018, 2019), quickly becoming the reference evaluation suite for natural language understanding. Both benchmarks were saturated, with models surpassing human-level performance within a few years of their introduction. Illustrating the capacity of competitions to drive rapid progress but also the challenge of keeping evaluation meaningful once the community converges on a solution. Big-Bench (Srivastava et al., 2022) and HELM (Liang et al., 2022) subsequently attempted to address saturation by broadening evaluation to hundreds of tasks and multiple evaluation axes (accuracy, calibration, robustness, fairness, efficiency), respectively.

### 4.1.2 SAFETY, ALIGNMENT, AND RED-TEAMING CHALLENGES

LLM competitions are paying explicit attention to safety and alignment. For instance, the Trojan Detection Challenge[61] at NeurIPS 2022 asked participants to identify backdoor triggers hidden in fine-tuned language models, targeting robustness concerns that had been difficult to study without a curated competition framework. Similarly, the ARC-Evals / METR[62] evaluation effort has motivated the community to design challenges that test not only capability but also the degree to which models are alingned with human values. On the other hand, red-teaming competitions, where participants attempt to elicit harmful, biased, or other undesirable outputs from a target model, have emerged as a crowd-sourcing mechanism for safety research. In this context, the Generative AI red teaming challenge[63] collocated with DEFCON aimed to identify failure modes in several frontier LLMs, producing a dataset of adversarial prompts that is now used in safety research. This form of competition is arguably a natural extension of the penetration-testing competitions long established in the cybersecurity community.

### 4.1.3 TEXT AND CODE GENERATION

The evaluation of free-form text generation has prompted novel competition designs. Notably, the ChatBot Arena (Chiang et al., 2024), introduced a crowd-sourced evaluation model in which human annotators rank the outputs of two anonymized models in pairwise comparisons. The resulting Elo-style leaderboard has become an influential reference for conversational LLM quality and has inspired similar arena-style evaluations for code generation and multimodal models. Coding challenges have proven to be a fruitful testbed for LLM evaluation. HumanEval (Chen et al., 2021) and the Mostly Basic Python Problems benchmark (Austin et al., 2021) established functional correctness as a primary metric, enabling automatic evaluation of free-form code produced by language models. SWE-bench (Jimenez et al., 2024) raised the bar further by posing real GitHub issues from open-source repositories as tasks, requiring models to produce patches that pass the associated test suites. These benchmarks have many of the structural properties of traditional competitions, a fixed task definition, a held-out evaluation set, and a public leaderboard, and have guided several generations of code-specialized LLMs.

---

61. https://trojandetection.ai/
62. https://metr.org/
63. https://humane-intelligence.org/get-involved/events/defcon-2023-overview/

Overall, the LLM era has brought both opportunities and challenges for the competition community. On one hand, it has democratized participation, while on the other hand it has raised new questions around evaluation validity, benchmark contamination, and the separation of genuine generalization from memorization of training data.

## 4.2 Challenges on AI Agents and Agentic AI

An autonomous agent is defined as an entity that perceives its environment, makes plans, and takes sequences of actions to achieve a goal. This notion has motivated competitions since the early days of RoboCup and the DARPA Grand Challenges (see Section 2.1). However, the convergence of LLMs with tool usage, memory, and multi-step planning has given rise to a new generation of *agentic AI* challenges that differ substantially from their predecessors in both scope and evaluation methodology.

### 4.2.1 Web and computer use agents

WebArena (Zhou et al., 2023) and WorkArena (Drouin et al., 2024) present agent benchmarks in which participants build systems capable of completing realistic web-browsing tasks in a sandboxed environment. The associated leaderboards have attracted considerable attention and revealed the significant gap between human performance and the best current agents. Mind2Web and VisualWebBench extend the evaluation to more diverse web environments and introduce vision as a primary input modality (Deng et al., 2023; Liu et al., 2024). OSWorld broadens the scope further by evaluating agents that interact with a full desktop operating system across a wide range of application software, constituting one of the most challenging open-ended agent benchmarks available (Xie et al., 2024).

### 4.2.2 Tool use and function calling

The evaluation of LLM tool usage has become a distinct sub-field within agent benchmarks. ToolBench evaluates whether an agent can correctly select and invoke tools from catalogs Xu et al. (2023), while BFCL (Berkeley Function Calling Leaderboard) provides a structured leaderboard for function-calling accuracy across programming languages and API formats (Patil et al., 2025). These benchmarks are particularly relevant for enterprise and production use cases where agents are expected to interact reliably with external systems.

### 4.2.3 Multi-agent challenges

An important dimension of agentic AI is the interaction between multiple agents, whether cooperative, competitive, or mixed. The Melting Pot benchmark evaluates the social generalisation of reinforcement learning agents across a diverse suite of multi-agent substrates (Leibo et al., 2021). Concordia extends this to LLM-based agents in social simulation settings (Vezhnevets et al., 2023). Multi-agent challenges introduce the additional complexity of emergent communication, coordination, and strategic reasoning, and several NeurIPS competition track entries have explored these themes (see Section 3.1). Cooperative multi-agent settings are of particular relevance for agentic AI given the growing

deployment of LLM-orchestrated pipelines in which multiple specialised agents collaborate to complete a task.

Summarizing, agent challenges represent one of the most rapidly evolving frontiers in academic competitions. They inherit and extend the tradition of interactive evaluation established by RoboCup and the DARPA series, while introducing new methodological demands around long-horizon evaluation, multi-modal perception, tool use, and safety. The connection to LLM challenges is intimate: most state-of-the-art agents today are built on top of foundation models, and progress on agent benchmarks is tightly coupled to progress in LLM capability and alignment.

### 4.3 Data-Centric Competitions

Historically, the dominant paradigm in machine learning competitions has been *model-centric*, that is, participants are given a fixed dataset and are asked to build the model that achieves the best score on a held-out test set. While effective, it has also been criticized for incentivizing marginal architectural improvements and hyper-parameter tuning at the expense of deeper scientific contributions, see Section 2.3.2. *Data-centric AI* inverts this relationship by fixing the model and asking participants to improve the dataset itself.

#### 4.3.1 Label quality and annotation

Dataperf a benchmark suite developed under the MLCommons umbrella, formalizes data-centric evaluation across multiple tasks including image classification, keyword spotting, and hate speech detection (Mazumder et al., 2023). Dataperf has introduced the notion of *data selection* challenges, where a budget on the number of training examples is imposed and participants must select the most informative subset. This directly implements active learning, core-set selection, and curriculum learning research within a competition framework.

#### 4.3.2 Dataset creation as a competition task

An emerging and particularly innovative competition format asks participants not merely to improve an existing dataset but to *create* a new one. The Dynabench [64] platform implements a human-and-model-in-the-loop evaluation paradigm in which annotators are asked to create examples that fool a current best model, and the resulting examples are added to an ever-growing dynamic benchmark. Competitions hosted on Dynabench for natural language inference, sentiment analysis, and question answering have produced datasets that are systematically harder than those assembled by conventional annotation pipelines, because adversarial human annotators deliberately probe model weaknesses.

Data-centric competitions are still maturing as a genre. Several open methodological questions remain: how to define a fair fixed-model baseline, how to prevent participants from effectively performing model selection under the guise of data selection, and how to credit contributions that improve generalisation on future test distributions rather than a fixed held-out set. Nevertheless, data-centric challenges represent an important complement to traditional model-centric competitions, and their emphasis on dataset quality is particularly

---

64. `https://dynabench.org/`

timely given the growing recognition that foundation models are only as good as the data on which they are trained.

## 5 Discussion

Academic challenges have been decisive for the consolidation of fields of knowledge. This chapter provided an historical review and an analysis of benefits and limitations of challenges, while it is true that competitions can have undesired effects, there is palpable evidence that they have boosted research across a number of fields. In fact there are several examples of breakthrough discoveries that have arosen in the context of academic competitions.

While we are witnessing the establishment of academic competitions as a way to advance the state of the art, the forthcoming years are promising. Specifically, we consider that the following lines of research will be decisive in the next few years:

- **Multimodal LLM challenges.** The integration of vision and language in modern foundation models has given rise to a new wave of multimodal challenges. However, challenges targeting heterogeneous an not common modalities (e.g., spectra, graphs, etc.) will emerge in the following years.

- **Multi-task and reasoning challenges.** Reasoning is a quality that is highly subjective to evaluate in LLMs, designing robust and highly reliable competitions for this topic represents a major opportunity.

- **Cooperative competitions.** Coopetitions is a form of crowd sourcing in which participants compete to build the best solution for a problem, but they cooperate with other participants in order to obtain an additional reward (e.g., information from other participants, higher scores, etc.).

- **Challenges for education.** Exploiting the full potential of challenges in education is a challenge itself, but we think this is a valuable resource for reaching wider audiences with assignments that require solving practical problems.

- **Academic challenges for good.** This is a topic being pursued and encouraged by evaluation forums and competition tracks, consider for instance the NeurIPS competition track (Escalante and Hadsell, 2019; Escalante and Hofmann, 2020; Kiela et al., 2022).

- **Dedicated publications for challenges.** There are few dedicated forums in which results of challenges are published (consider for instance the Challenges in Machine Learning series[65]). We foresee more dedicated venues will be available in the next few years.

---

65. https://www.springer.com/series/15602

## Acknowledgments and Disclosure of Funding

The authors are grateful with Isabelle Guyon and Adrien Pavao for comments and suggestions that improved the manuscript.

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
