# OpenReview forum: "Academic Competitions"
_DMLR — Accepted by DMLR_

### Review · Reviewer_pQeX · 2024-09-09

**Recommendation:** 4
**Confidence:** 2

**Summary Of Contributions:**

This paper provides a thoughtful review of academic competitions and its role in driving forward research progress. The paper provides many references and pointers to recent academic competitions, including many well-known ones such as KDD cups and NeurIPS competitions as well as lesser-known ones. This writing can facilitate further discussions around the role of academic competitions in serving the research community.

**Strengths:**

I find the paper to be well-written and the paper discusses a topic that has been less thought about, so I think this writing contributes to the discussion and is a good add to the literature.

Another notable aspect of this paper is its focused discussions on each individual areas such as CV, NLP, transportation, etc.

**Audience:**

Yes

**Claims And Evidence:**

This paper discusses the nature of academic competitions. I find the discussion to be grounded in facts and clear arguments.

**Datasets And Benchmarks:**

N/A

**Extended Submissions:**

N/A

**Limitations:**

Some proofreading would be required before the paper is published.

**Requested Changes:**

I'm not sure if there're any other surveys of this nature. If so, I think it would be good to discuss those surveys in this context; if not, perhaps this could be stated more explicitly so that readers don't have to guess.

Another request is that it may be useful if the links/data repositories could be gathered together in some kind of online repository. This could be a useful reference point for researchers interested in looking into the academic competitions.

**Strengths And Weaknesses:**

This paper is generally well-written and the survey is comprehensive in its scope.

There are various typos such as mismatched formats across paragraphs and section headers, which can be fixed with some proofreading.

---

### Review · Reviewer_VVCZ · 2024-09-15

**Recommendation:** 3
**Confidence:** 3

**Summary Of Contributions:**

This chapter covers the development of academic competitions in machine learning and, in general, artificial intelligence. From centuries till today, this chapter provides a comprehensive introduction about

1. history of academic challenge,
2. the pros and cons of academic challenge,
3. the key factors of designing a good academic challenge,
4. the detailed discussion about challenges by subfields.
5. the future of academic challenge.

**Strengths:**

See above.

**Audience:**

Yes

**Claims And Evidence:**

NA

**Datasets And Benchmarks:**

This is a special submission.

**Extended Submissions:**

NA

**Limitations:**

Please see the Requested changes above.

**Requested Changes:**

I am not sure whether this is appropriate, given that this draft seems to be finished before the boom of LLMs. But for reference purposes, I would like to suggest some discussion on competitions about LLMs for the following reasons:

1. Competitions about LLMs are nowadays very important in both media propaganda and the development of LLMs.
2. The competition about LLMs takes various forms. It is not only a benchmark for participants to achieve higher scores but also some arenas (e.g. https://lmarena.ai/) for LLMs to compete with each other.
3. The benchmarks for LLMs are also sometimes criticized for not reflecting their true strength. I think there might be a lot of new reasons to discuss this.

**Strengths And Weaknesses:**

Strength:

1. the authors paid great attention to the existing competition and, therefore, provided a compressive survey.
2. the argument made by the authors about how to develop an excellent academic challenge is inspiring.

Weakness:

1. Regarding the outlook for the next few years, I think the impact of LLMs and competitions for LLMs could also be an essential aspect to discuss.

---

### Review · Reviewer_KXGd · 2026-04-03

**Recommendation:** 3
**Confidence:** 2

**Summary Of Contributions:**

This paper provides a practical roadmap for designing and organizing ML/AI competitions. It is structured around four pillars (good problem, proposal review, good logistics, good participation) and covers the full lifecycle from conception to post-challenge analysis.

**Strengths:**

See above.

**Audience:**

Yes

**Claims And Evidence:**

The paper is experience-driven and guideline-oriented rather than empirical. The claims are well-supported by practical examples and references.

**Datasets And Benchmarks:**

This is a special submission (book chapter).

**Extended Submissions:**

This is a special submission (book chapter).

**Limitations:**

The paper is written almost entirely from the organizer's perspective. A brief acknowledgment of known downsides of competitions (e.g., incentivizing engineering over scientific insight, reproducibility challenges) would provide a more balanced view.

**Requested Changes:**

Add discussion on how LLMs have reshaped competition design — specifically arena-style evaluation, data contamination risks, and benchmark saturation. These are not minor trends and are directly relevant to the chapter's core topics (metrics, data leakage, protocol design). This would strengthen the chapter but is not strictly critical for acceptance. Proofreading for typos (e.g., "Historcal" in Table 1) should be done before publication.

**Strengths And Weaknesses:**

Strengths: Highly practical guide grounded in the authors' extensive experience. The protocol hierarchy, difficulty calibration discussion, and data leakage taxonomy are all useful contributions. The inclusion of a real proposal example makes the guidelines concrete.
Weaknesses: Limited coverage of recent shifts in competition design driven by LLMs (arena-style evaluation, benchmark contamination, saturation). Appendix A is disproportionately long. Minor formatting issues and typos remain.